# Eugenol; Effective Anthelmintic Compound against Foodborne Parasite *Trichinella spiralis* Muscle Larvae and Adult

**DOI:** 10.3390/pathogens12010127

**Published:** 2023-01-12

**Authors:** Mai ElGhannam, Yasser Dar, Mostafa Hassan ElMehlawy, Fatma A. Mokhtar, Lamia Bakr

**Affiliations:** 1Zoology Department, Faculty of Science, Tanta University, Tanta 31527, Egypt; 2Department of Pharmacognosy, Faculty of Pharmacy, Al Salam University, Kafr El-Zayat 31611, Egypt

**Keywords:** *Trichinella spiralis*, eugenol, anthelminthic, albendazole

## Abstract

Trichinosis is a foodborne parasitic infection that results from ingestion of raw or under-cooked pork meat infected by parasitic nematode *Trichinella spiralis* with cosmopolitan distribution. Anthelmintic drugs are used to eliminate intestinal adult parasites and larvae as well as tissue-migrating newborn and in-turn encysted larvae. However, eliminating the infection or averting it from transmission is rarely possible using anthelmintic groups of benzimidazole derivatives. Eugenol (EO) is the main extracted constituent of clove oil (80–90%) and is responsible for its aroma. Therefore, this study aims to investigate the effect of eugenol on both adult and muscle larvae of *Trichinella spiralis* in vitro. IC50 for different concentrations of eugenol were calculated for both muscle larvae (187.5 µM) and adults (190.4 µM) to determine the accurate dose range. Both the nematode stages were cultured in the commonly used RPMI-1640 media in 24-well plates. Different concentrations of eugenol (122, 305, 609, 1218, and 3045 µM) were administered in different groups of larvae/adults. The parasitological parameters were monitored after 1, 3, 6, 10, 24 h for each EO concentration in concomitant with the control groups. Reference chemotherapeutic anthelminthic drug “albendazole” (at dose 377 µM) was experimentally grouped in triplicates as positive control and the untreated as negative control, respectively. Mortality was observed where time-dependent adult stages were less susceptible than muscle larvae. Eugenol achieved 100% efficacy against *T. spiralis* larvae and killed the total larvae after 10 and 24 h at concentrations of 1218 and 3045 µM, the same as albendazole’s effect on the positive control group. In regard to adults, resembling muscle larvae (ML), a significant effect of both concentrations at *p* < 0.0001 was obtained, and the concentration × time interaction was significant at *p* < 0.0001. Furthermore, the treated/untreated adult and muscle larvae were collected and processed for scanning electron microscopy (SEM). Massive destruction of parasite burden was observed, especially at high concentrations (1218 and 3045 µM). In addition, complete and mild loss in cuticular striation in both the treated and positive controls were confirmed by SEM, respectively, in comparison to the control untreated group.

## 1. Introduction

Trichinosis is a zoonosis induced by the parasitic genus *Trichinella*. Humans are infected through the consumption of raw or undercooked meat containing infective larva. Although this event occurs accidentally, the global prevalence of trichinosis is estimated to be around 11 million in humans [1]. *Trichinella spiralis* is the most virulent and pathogenic species to humans and has two lifecycle phases: an intestinal phase and a muscular phase [1]. In the enteral phase, muscle tissues are digested in the stomach, and larvae are released; larvae penetrate the intestinal mucosa of the small intestine and reach the adult stage within 48 h post-infection (p.i.) and mate; the female worm releases newborn larvae in the lymphatic vessels (from the fifth day p.i. onwards; the length of newborn production, from 1 week to several weeks, is influenced by host immunity) [1,2].

In the parenteral phase, the newborn larvae reach the striated muscle and actively penetrate in the muscle cell; the larva grow to the infective stage in the nurse cell (the former muscle cell); and, after a period of time (weeks, months, or years), a calcification process occurs. Humans can become infected by ingesting infective *T. spiralis* larvae in undercooked pork meat [1].

Human trichinosis is commonly treated with benzimidazole derivatives such as albendazole, mebendazole, flubendazole, and thiabendazole. But none of these drugs is completely effective to kill encapsulated larvae and newborn larvae (NBL) because of their low bioavailability [1,2]. Furthermore, when they were used at high doses, some side-effects could be seen. In addition, most of them are contraindicated in pregnant women and in children who are under 2 years old [2]. For these reasons, an effective therapy must be improved, and new drugs are needed. Some plant extracts such as *Artemisia vulgaris, Artemisia absinthium*, garlic extract, myrrh and thyme extracts, *Nigella sativa, Allium cepa*, and *Mangifera indica* were reported to possess anthelmintic activities [3,4,5,6,7]. Nowadays, scientists show increased interest in producing anthelmintic pharmaceutical formulations from plant origin [8,9].

Eugenol, 4-allyl-2-methoxyphenol, is a phenylpropanoid with an allyl chain-substituted guaiacol. Eugenol, a naturally occurring compound, has been reported to be present in several plant families including Holy basil or tulsi leaves (*Lamiaceae*), *Eugenia caryophyllata* (clove) [10], *Zingiber officinale* (ginger) [11], bark and leaves of *Cinnamomum verum* (cinnamon) [12], *Curcuma longa* (turmeric) [13], as well as various aromatic plants such as *Ocimum basilicum* (basil) [14], *Myristica fragrans* (Houtt.) (nutmeg) [15], and *Cinnamomum loureiroi* Nees. (Saigon cinnamon) [10]. The major natural sources of eugenol are *Eugenia caryophyllata* (syn *Syzygium aromaticum*), which comprises 45–90% [16], and cinnamon, which contains 20–50% of eugenol; however, the commercial level extraction of eugenol is quite expensive with longer cultivation times, while ginger and tulsi can be used instead of cinnamon and clove as cheaper sources [13].

Eugenol bears huge industrial applications, particularly in pharmaceutics, dentistry, flavoring of foods, agriculture, and cosmetics because the World Health Organization (WHO) has declared eugenol as a nonmutant and generally recognized as safe (GRAS) molecule. Activities of eugenol show remarkable anti-inflammatory, antioxidant, analgesic, antifungal, and antimicrobial properties and have a significant effect on human health. It has fungicidal, anticarcinogenic, antiallergic, antimutagenic, and insecticidal properties as well [17].

In the parasitological field, the antiparasitic activity of eugenol was tested and showed anti-inflammatory and antifibrotic effects against *schistosoma mansoni* [18] using a dose of 500 ug/kg/day. Certain drastic effects of eugenol have also been reported on morphology and growth of various parasites such as *Trypanosoma cruzi* [19], *Giardia lamblia* [20], and *Leishmania donovani* [21]

To date, the anthelminthic properties of eugenol have not been well studied. Hence, the present work aims to evaluate the effect of eugenol on the viability of *Trichinella spiralis* life stages (adults and muscle larvae) as an anthelmintic drug in vitro for the first time.

## 2. Materials and Methods

### 2.1. Parasites

The *T. spiralis* strain was obtained as a muscle larvae rat (donor) from the Parasitology Department of the Biological unit of Theodor Bilharz Research Institute, Imbaba, Giza, Egypt (TBRI), where all approved research work complied with the World Medical Association of Ethics under Federal Wide Assurance No. FWA00010609. Swiss albino mice were kept in equal laboratory amendment of standard commercial pelleted diet with freely accessible water and ensuring good sanitary conditions throughout the time of the study. Upon arrival, all experimental animals were left for one week to be quarantined for any illness and to be acclimatized to the conditions of laboratory environment. Mice received standard diet and chlorine-free tap water and were orally infected with 200 *T. spiralis* larvae each. [22,23]. All care and procedures adopted for the present investigation were in accordance with the approval of the Institutional Animal Ethics Committee of Zoology Department, Faculty of Science, Tanta University with ethical protocol number IACUC-SCI-TU-0042.

### 2.2. Isolation of T. spiralis Adult Worms and Muscle Larvae

According to the previous study by Ozkoc et al. [24], adults and muscle larvae of *Trichinella spiralis* were obtained from the infected mice. Briefly, Swiss albino mice infected with *T. spiralis* for 30 days were sacrificed. [25] *Trichinella spiralis* adult worms were isolated from the small intestines of infected untreated mice six days post-infection (P.I.). Larval muscles were recovered from the carcasses of infected mice after 30 days post infection by incubating minced skinned mouse with artificial digestive fluid (1% pepsin, *w*/*v* & 1% HCL, *v*/*v*) in a conical flask at 37 °C overnight. They were prepared and counted according to Wakelin and Wilson [26]

### 2.3. Drugs

Albendazole was purchased as 400 mg/10 mL oral suspension from Pharma Cure Pharmaceutical Industries, Egypt, whereas eugenol was purchased from Fuzhou Farwell Import & Export Co., Ltd. (Hong Kong, China). 

### 2.4. In Vitro Experimental Design

Out of total number of 525 adult and 2100 muscle larvae (ML) *T. spiralis*, only 25 adults and 100 larvae per well, respectively, were cultured in a 24-well tissue culture plate prepared with an incubation medium consisting of Rapid Prototyping and Manufacturing Institute (RPMI-1640) medium (containing 10% fetal calf serum, 200 U/mL penicillin and 200 μg/mL streptomycin). 

Three groups were established in this study:Group I—muscle larvae and adult worms cultured in the incubation medium only.Group II—muscle larvae and adult worms cultured in the incubation medium containing eugenol that was at concentrations of (122, 305, 609, 1218, and 3045 µM).Group III—muscle larvae and adult worms cultured in the incubation medium containing albendazole that was at a concentration of 377 µM [27].

Tests were performed in 3 replicates, and the plate was placed in the incubator at 37 °C and 5% carbon dioxide for 1, 3, 6, 10, and 24 h. Then, all adult worms and larvae (both dead and living) were counted, collected, and processed for scanning electron microscopic study.

In the present study, parasite viability was performed by counting manually. To calculate the viability rates of parasites, definitely dead parasites were counted. Even if the parasites had very little motility, they were defined as alive.

### 2.5. Scanning Electron Microscopy (SEM)

The effect of eugenol on ML and adult of *Trichinella spiralis* is also supported by scanning electron microscopy. After 24 h of incubation, *T. spiralis* worms and larvae of each group of in vitro experiment were removed and fixed by immersing them immediately in 4E1G (fixative, phosphate buffer solution) PH = 7.4 at 4 °C for 3 h. Specimens were then postfixed in 2% OsO_4_ in the same buffer at 4 °C for 2 h. Samples of *Trichinella spiralis* (adult and larvae) were dried by means of critical point method, mounted using an AL-stub and coated with gold up to a thickness of 400 A in a sputter-coating unit (JFC-1100E) [28].

### 2.6. Statistical Analysis

Two-way repeated-measures analysis of variance (ANOVA) (concentration × time), followed by the Tukey’s multiple comparisons test, was used to analyze group data. GraphPad Prism-7 was used to perform the statistical analyses in this study. Data were presented as the mean ± standard deviation (SD) and the significance of the differences between experimental and control groups were analyzed by Multiple *t*-test. Statistical significance was defined as *p* < 0.05.

## 3. Results

### 3.1. IC50 Calculation 

IC50 for different concentrations of eugenol was calculated for both muscle larvae (187.5 µM) and adults (190.4 µM) (Figure 1 and Figure 2).

### 3.2. Parasitological Studies 

#### 3.2.1. Muscle Larvae Count

In the current study, the effect of eugenol on the viability of *T. spiralis* muscular larvae in relation to albendazole was evaluated. Eugenol reduced the viability of *T. spiralis* larvae in a time and dose dependent manner; eugenol achieved 100% efficacy against *T. spiralis* larvae and killed the total larvae after 10 and 24 h at concentrations of 1218 and 3045 µM, respectively, whereas at concentration of 609 µM, total death occurred after only 24 h. The larvicidal effect in different concentrations of eugenol was significantly different (*p* < 0.05) compared to the control group. The viability rates of larvae decreased with prolonged exposure at lower concentrations (Table 1). As compared with the control, at 122, 305, and 609 µM concentrations, the effects were significantly higher from 3 to 24 h than those at 1 hr (*p* < 0.05). In addition, albendazole killed all the larvae after 10 and 24 h of incubation and this effect was significantly different (*p* < 0.05) compared to the control group (Figure 3).

#### 3.2.2. Adult Worm Count

The lethal effect of eugenol on the adult form was dose- and time-dependent, similar to the ML (Table 2). At 1218 and 3045 µM concentrations, wormicidal effect was observed only after 24 h. However, at concentration of 3045 µM, this effect only begun after 10 h of incubation (*p* < 0.05). As compared with controls at 305 and 609 µM concentrations, the effects were significantly higher at 3–24 h than those at 1 h (*p* < 0.05). With the 122 µM concentration, this effect began after 10 h of exposure (*p* < 0.05). Albendazole also showed wormicidal effect after 24 h, which was significantly different in comparison to the control (*p* < 0.05) (Figure 4).

A significant effect of both concentration [F (6, 14) = 386.6, *p* < 0.0001] and time [F (4, 56) = 847.1, *p* < 0.0001] was found, in addition the concentration × time interaction was considerable [F (24, 56) = 47.36, *p* < 0.0001]. Tukey’s multiple comparisons test revealed a significant reduction of ML at conc. of 3045 µM in all tested hours compared to the control group, whereas there was no significance revealed between 3045 µM and albendazole groups at different hours of incubation except after 3 h incubation time (*p* = 0.0015).

Concerning adults that resemble ML, a significant effect of both concentration [F (6, 14) = 107, *p* < 0.0001] and time [F (4, 56) = 743.2, *p* < 0.0001] was found, and the concentration × time interaction was also considerable [F (24, 56) = 28.46, *p* < 0.0001]. Significant reduction of adult worm was revealed, by Tukey’s multiple comparisons test, at conc. of 3045 µM in all tested hours compared to the control group. Tukey’s test also indicated no significant difference between 3045 µM and albendazole groups at different hours of incubation, except after 1 h of incubation time (*p* = 0.0288).

#### 3.2.3. SEM (Scanning Electron Microscopy) Findings

Regarding the infected control group, when cultured in the incubation medium only, the cuticle of the larvae and adult worm retained the normal structure in the form of ridges, transverse creases, and annulations (Figure 5A), with the appearance of openings of the hypodermal gland in adults (Figure 6B). In albendazole-treated groups, there was severe destruction of larvae in the form of marked mild smoothing of cuticle annulation and multiple fissures in the cuticle (Figure 5B). In the eugenol-treated group, there was severe destruction of larvae, with complete smoothing of annulation and presence of multiple blebs and vesicles of the cuticle (Figure 5C).

As for the adults, in the albendazole-treated groups, there was severe destruction of adult worm, showing swelling and mild smoothing of the cuticle annulation (Figure 6B). While in the eugenol-treated group, there was a complete smoothing of the cuticle annulation and presence of multiple cuticular blebs and fissures (Figure 6C)

## 4. Discussion

The medical treatment of trichinosis is a matter of much debate. Albendazole, one of the benzimidazoles, is still the available drug of choice in trichinosis treatment. However, it is reported that albendazole can cause multiple serious systemic adverse drug reactions, such as severe drug eruptions, encephalitis, epilepsy, and even death [2,29]. Moreover, it shows poor susceptibility to migrating and encapsulated muscle larvae [30]. These data elucidate the urgent need for a new, safe, and effective treatment capable of eradicating the *Trichinella* spp. infection.

In the present study, the anthelmintic activity of eugenol oil against *Trichinella* spp. was tested in vitro. To the best of our knowledge, this is the first report to evaluate this plant extract activity against *Trichinella* spp. Tukey’s multiple comparisons test of ML groups revealed a significant effect of both concentration [F (6, 14) = 386.6, *p* < 0.0001] and time [F (4, 56) = 847.1, *p* < 0.0001]; in addition, the concentration × time interaction was significant [F (24, 56) = 47.36, *p* < 0.0001]. Concerning adults, resembling ML, a significant effect of both concentration [F (6, 14) = 107, *p* < 0.0001] and time [F (4, 56) = 743.2, *p* < 0.0001] was found, and the concentration × time interaction was significant [F (24, 56) = 28.46, *p* < 0.0001] as well. Thus, the efficacy of eugenol on both *T. spiralis* ML and adults is dose–time dependent.

Eugenol has lethal activity against both of ML and adult stages of *T. spiralis.* This remarkable effect was found to occur in a dose and time dependent manner. Although adult stages were less susceptible than ML, no viable adults were seen at 3045 µM concentration at higher incubation periods (same as the albendazole-treated group—reference drug). However, in the lowest eugenol concentration (122 µM), there was no significant difference in viability of adults at lower incubation periods while a significant reduction was observed in ML within 3 h (*p* < 0.05).

The significant lethal effect of eugenol on the ML was also shown in the highest concentrations of (1218 and 3045 µM, respectively) >6 h, similar to the albendazole-treated group—reference drug. In addition, the viability of the ML was interestingly higher in the control than in (122 µM) eugenol concentration after 24 h incubation. Appearance of dead parasites in a short period, especially at high concentrations, shows the possible direct effect of eugenol on the parasite. These results are consistent with previous research on eugenol emulsion against leishmaniasis infection found that a Th1 immunostimulant-polarizing mechanism increased antileishmanial activity of eugenol without producing any adverse effects [22]. The present result also agreed with the analogues study on the in vitro effect of resveratrol (natural phytoalexin form mainly in grapes) against larvae and adults of *T. spiralis* and found that resveratrol has antiparasitic activity on larval and adult stages of *T. spiralis* in a dose–time-dependent manner [24].

Results obtained herein agree with previous study [31], which reported that eugenol and methyl isoeugenol exhibited an inhibitory effect at higher concentrations against infective larvae of four of major ovine gastrointestinal nematodes *Haemonchus contortus, Trichostrongylus axei*, *Teladorsagia circumcincta*, and *Trichostrongylus vitrinus*. This is in agreement with [32], which reported that myrrh either as crude extracts or volatile oil had promising in vitro larvicidal activities against *T. spiralis* larvae in comparison with albendazole in a dose–time-dependent manner. 

In the current study, the electron microscopy scans showed severe destruction of both larvae and adult worm, marked cuticle swelling, areas with fissures and blebs, and severe and mild smoothing of annulations at higher concentrations of eugenol- and albendazole-treated groups, respectively. SEM results agreed with a former study on myrrh where *T. spiralis* muscle larvae treated with crude myrrh extract caused extensive cuticle damage [32]. The findings additionally agree with results that proved the ability of Graviola extract to destruct the adult worm of *T. spiralis’* cuticle, accompanied by cuticle swelling and vesicles formation, blebbing, as well as loss of annulations [33]. This coincides with a previous study wherein marked destruction of the cuticle of *T. spiralis* adult treated with chitosan loaded to albendazole was observed [34]. As the cuticle is essential for osmoregulation and is the main route of implemented drugs’ passage into nematodes in transcuticular passive diffusion with subsequent destruction of the worm’s surface [35]. All these findings indicate effectiveness of the antiparasitic activity of eugenol oil as a proposed natural anthelmintic drug. 

## 5. Conclusions 

As far as we know, this is the first report presenting the results of anthelmintic activity of eugenol against *T. spiralis*. Eugenol had a remarkable lethal effect on ML and adult forms in a dose–time-dependent manner. This significant lethal effect of eugenol on the adult and ML was shown in the highest concentrations of 1218 and 3045 µM at higher incubation time. These results were confirmed by SEM in the form of complete smoothing of the cuticle annulation, extensive cuticle damage, and blebbing. These results suggest that eugenol might potentially have therapeutic value in the early stages of human trichinosis.

## Figures and Tables

**Figure 1 pathogens-12-00127-f001:**
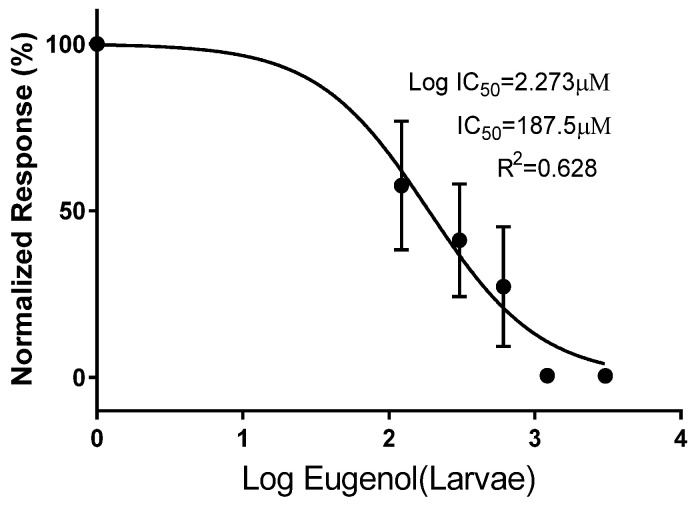
In vitro IC50 (half inhibitory concentration response) of different concentrations of eugenol on *Trichinella spiralis* ML.

**Figure 2 pathogens-12-00127-f002:**
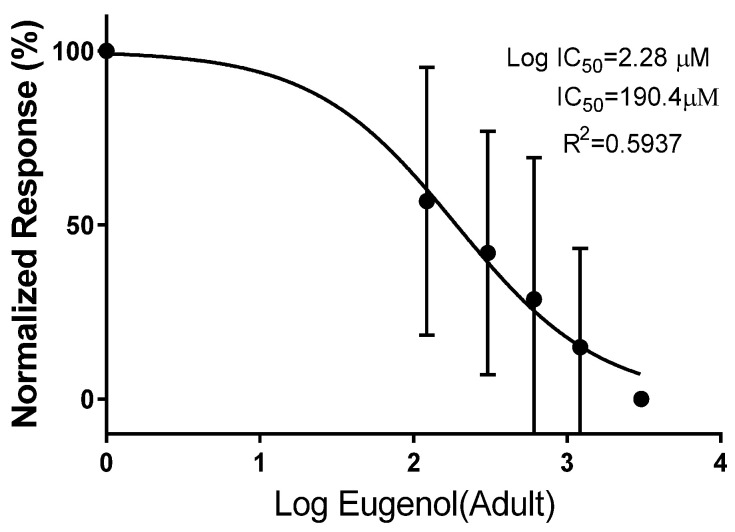
In vitro IC50 (half inhibitory concentration response) of different concentrations of eugenol on adult *Trichinella spiralis*.

**Figure 3 pathogens-12-00127-f003:**
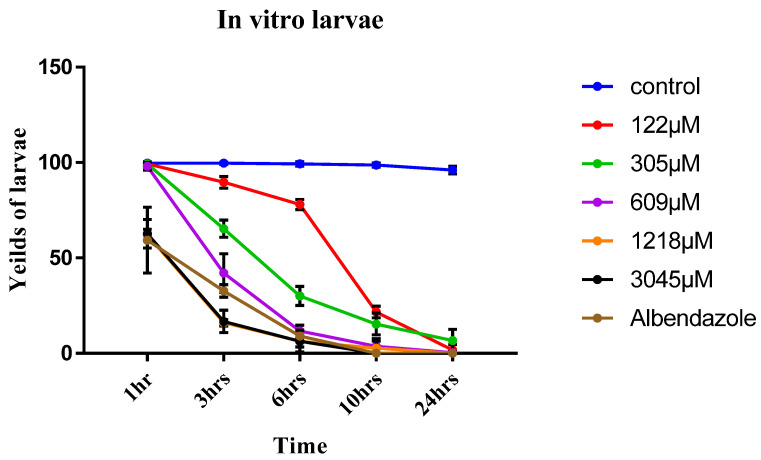
In vitro yields of *Trichinella spiralis* ML at different concentrations of eugenol (122, 305, 609, 1218, and 3045 µM) and different times (1, 3, 6, 10, and 24 h) using albendazole as a reference drug indicates significant differences vs control group at *p* ≤ 0.05 determined using *t*-test.

**Figure 4 pathogens-12-00127-f004:**
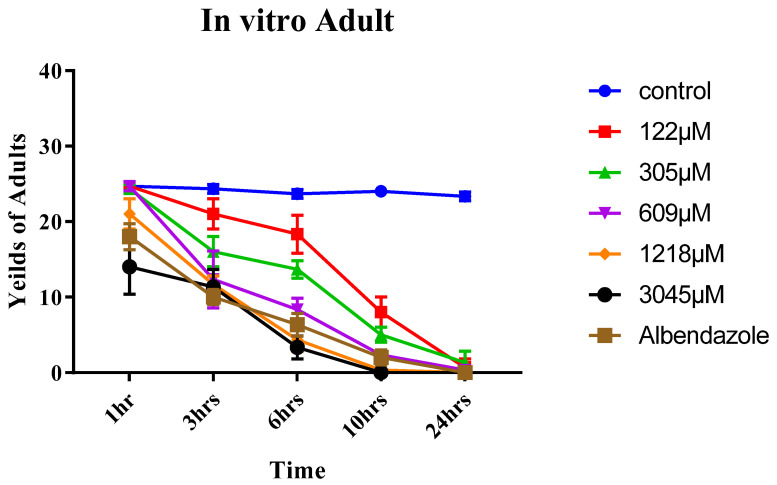
In vitro yields of adult *Trichinella spiralis* at different concentrations of eugenol (122, 305, 609, 1218, and 3045 µM) and different times (1, 3, 6, 10, and 24 h) using albendazole as a reference drug indicate significant differences vs. control group at *p* ≤ 0.05 determined using *t*-test.

**Figure 5 pathogens-12-00127-f005:**
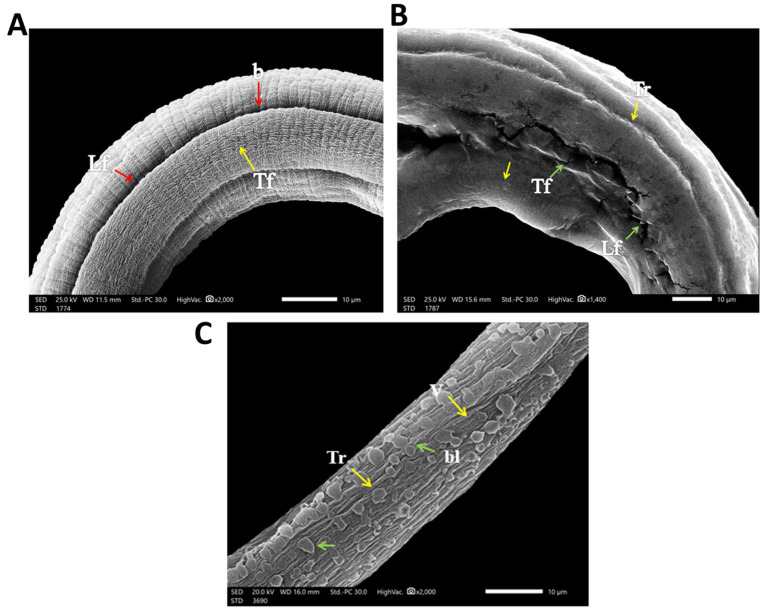
SEM findings of cultured *T. spiralis* larvae. (**A**) Normal cultured *T. spiralis* larva trunk without any effect of any drugs or chemicals showing the normal architecture of the cuticle annulation or transverse fissures (Tf) (yellow arrows) and longitudinal fissures (Lf) with the presence of bacillary openings (b) (red arrows). (**B**) Albendazole-treated cultured *T. spiralis* larva trunk showing mild smoothing of the cuticle annulation and transverse ridges (Tr) (yellow arrows) with the presence of multiple longitudinal fissures (Lf) and transverse fissures (Tf) of the cuticle (green arrows). (**C**) Eugenol-treated cultured *T. spiralis* larva trunk showing complete loss and deformities in the cuticle annulation and transverse ridges (Tr) (yellow arrows) and an area with multiple blebs (bl) and vesicles (V) of the cuticle (green arrows).

**Figure 6 pathogens-12-00127-f006:**
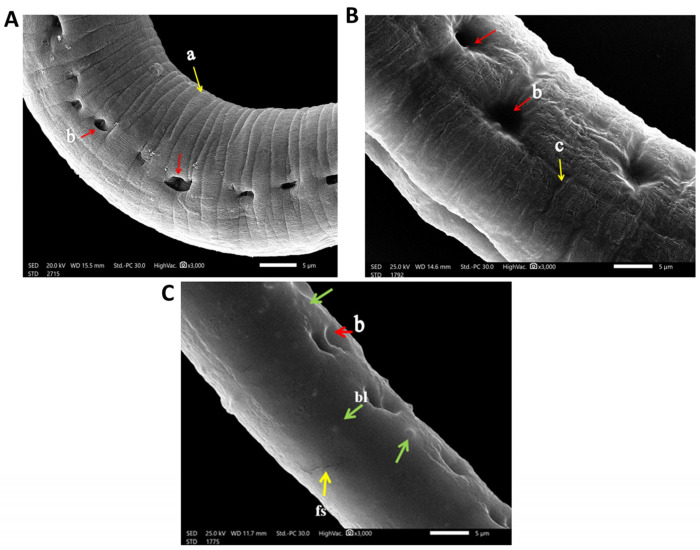
SEM findings of cultured adult *T. spiralis.* (**A**) Control untreated adult *T. spiralis* trunk showing the normal architecture of the cuticle annulation and transverse ridges) (a) (yellow arrows) with the presence of bacillary bands glands openings (b) (red arrows). (**B**) Albendazole-treated cultured adult *T. spiralis* trunk showing swelling, mild wrinkling and smoothing of the cuticle annulation (c) (yellow arrow) as well as wideness in openings of bacillary bands glands (b) (red arrows). (**C**) Eugenol-treated cultured adult *T. spiralis* trunk showing complete smoothing of the cuticle annulation, presence of mild wide openings of bacillary band glands (b) (red arrows) and area with multiple blebs (bl) (green arrows) and fissures (fs) (yellow arrows), respectively.

**Table 1 pathogens-12-00127-t001:** In vitro effect of eugenol on *T. spiralis* muscle larvae in relation to time of exposure.

Eugenol Dose		Time (h/s)			
	1 h	3 h	6 h	10 h	24 h
Control	99.667 ± 0.577	99.667 ± 0.577	99.333 ± 1.155	98.667 ± 1.155	96 ± 2
122 µM	99.333 ± 1.155	89.667 ± 3.055 *	78 ± 2.646 *	21.667 ± 3.055 *	1.667± 2.887 *
305 µM	99.333 ± 1.156	65.333 ± 4.509 *	30 ± 5 *	15.333 ± 5.686 *	6.667 ± 5.859 *
609 µM	98 ± 2	42 ± 10.149 *	11.667 ± 3.055 *	3.667 ± 4.0415 *	0.333 ± 0.577 *
1218 µM	62 ± 3 *	16 ± 1.732 *	6.333 ± 3.055 *	2.667 ± 3.786 *	0
3045 µM	62.667 ± 7.506 *	16.667 ± 5.859 *	6.333 ± 5.686 *	0	0
Albendazole	59.333 ± 17.214 *	32.667 ± 3.215 *	9 ± 3 *	0	0

Data expressed as mean ± SD; * *p* < 0.05, significantly different compared with control group determined using multiple *t*-test.

**Table 2 pathogens-12-00127-t002:** In vitro effect of eugenol on adult *T. spiralis*.

Eugenol Dose		Time (h/s)			
	1 h	3 h	6 h	10 h	24 h
Control	24.667 ± 0.577	24.333 ± 0.577	23.667 ± 0.577	24 ± 0	23.333 ± 0.577
122 µM	24.667 ± 0.578	21 ± 2	18.333 ± 2.517	8 ± 2 *	0.667 ± 1.155 *
305 µM	24.333 ± 0.577	16 ± 2 *	13.667 ± 1.155 *	5 ± 1 *	1.333 ± 1.528 *
609 µM	24.667 ± 0.577	12.333 ± 3.786 *	8.333 ± 1.528 *	2.333 ± 0.577 *	0.333 ± 0.577 *
1218 µM	21 ± 2 *	11.667 ± 1.155 *	4.333 ± 1.528 *	0.333 ± 0.577 *	0
3045 µM	14 ± 3.606 *	11.333 ± 2.309 *	3.333 ± 1.528 *	0	0
Albendazole	18 ± 1.732 *	10 ± 1 *	6.333 ± 1.528 *	2 ± 1 *	0

Data expressed as mean ± SD; * *p* < 0.05, significantly different compared with control group determined using multiple *t*-test.

## Data Availability

Not applicable.

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
