# Peer review of "Eugenol; Effective Anthelmintic Compound against Foodborne Parasite Trichinella spiralis Muscle Larvae and Adult"

_pathogens, 2023, doi:10.3390/pathogens12010127_

Round 1

Author Response

  1. Why do you select eugenol for this anthelmintic study?

This is mentioned in manuscript in the section of introduction  (lines  40-46) and (lines 62-72).we highlighted it in yellow colors.

In parasitological field, the anti-parasitic activity of eugenol was tested and showed anti-inflammatory and antifibrotic effects against schistosoma mansoni and certain drastic effects of eugenol have been also reported on morphology and growth of various parasites like Trypanosoma cruzi, Giardia lamblia, and Leishmania donovani. So,eugenol selected for this anthelmintic study.

  1. Eugenol 25 at different concentrations (500, 200, 100, 50 and 20 μg/mL) were used for

this study. How you decided this dose range?

we decided this dose range according to the IC50   adjustment of different doses resulting in this curve for each of muscle larvae and adult worms and found those doses better for the experiment. And this is mentioned and illustrated in materials and method section as figurec 1 and figure 2  and the dose range was compatable with previous study

 Eugenol, a potential schistosomicidal agent with anti-inflammatory and antifibrotic effects against schistosoma mansoni, induced liver pathology. Infection and drug resistance 2019, 12, 709.

  1. Many places in this manuscript scientific name is not in Italis, Trichinella spiralis.

Please correct it.

The whole paper is checked and corrected

  1. Lane 229-232, “Significant reduction of adult worm at conc. 50% in all tested hours

compared to the control group...”, please simply the sentence with clarity.

We simplified the sentence

We mean that the best effect of eugenol revealed at conc 50% of eugenol and this efficacy observed at all tested hours (1,3,6,9,10and 24hrs)in comparisomn to the control negative group  while there is non-significance revealed between conc50% and albendazole groups at different hours of incubation (showing the same lethal effect except at 1hrs of incubation time whereas (P= 0.0288 ).these results are obtained by Tukey’s multiple comparisons test.

  1. In the lane 321, “Results obtained herein agree with previous study of [36] who

reported that...” instead of who specific name of author or rewrite the sentence without

out who...

we corrected the discussion section, and rewrite the sentence without who

  1. In discussion, please specify any possible mechanism of action of eugenol on this

context based on your in vitro and bio information studies.

According to other reviewer opinion, we removed the bioinformatic study. But concerning the in vitro study, this illustrated  in discussion section from line  251 to line 263 revealing the efficacy of eugenol on both T. spiralis adult and ML in dose-time dependent manner. Eugenol oil has lethal activity against ML and adult stages of T. spiralis in vitro.This remarkable effect was found to occur dose and time dependent manner. Although adult stages were less susceptible than ML, the total mortality was seen at 50% concentration at higher incubation periods (after 10 and 24 hrs incubation time) similar to albendazole treated group –reference drug-. This is in agreeing with previous studies in reference [1], [24], [36] and [37].

  1. Also, you mentioned about the target genes ALOX15, HDAC8, PTGS1 and VEGFA.

Specify their anthelmintic properties, in brief

According to other reviewer opinion, we removed the bioinformatics study at all

  1. In conclusion, you should point out to the main finding obtained from your research

Work

Following your scientific advice, we added the main finding in the conclusion section from line 298 to line 302 and highlighted it with yellow color.

  1. In future, Do more specific gene expression studies to confirm eugenol’s mechanism

of action

We intend to do a bioinformatic big study accompanied with gene expression in the future and pathway analysis in our future study concerning the anthelmintic properties of eugenol.

Reviewer 2 Report

It should be remembered that the very structure and characteristics of terpenes, their lipophilicity, disrupts the integrity of cell membranes, disrupts the functioning of proton pumps, therefore, without experimental analysis, gene expression studies, modeling we couldn’t say about influence and target genes.

Manuscript as only parasitological experimental data without bioinformatics, properly conducted, summarized, and discussed would be possible to consider in this form not suitable for publication. So much work put in for nothing, it's a shame.

Author Response

The manuscript was incorrectly and carelessly written. This study analyzed the influence of essential oil (eugenol) to larval stages of Trichinella during in vitro cultivation with depend on time and concentration. It presents an extensive collection of data on the modulation of parasitological maker after 1,3,6,10, and 24 hours of eugenol exposure.

It should be remembered that the very structure and characteristics of terpenes, their lipophilicity, disrupts the integrity of cell membranes, disrupts the functioning of proton pumps, therefore, without experimental analysis, gene expression studies, modeling we couldn’t say about influence and target genes.

The parasitological methods used are good but bioinformatic is not worthy of a scientific achievement at all. The publication represents an experiment and the objectives assumed by the authors characteristic chaos and mess.

Manuscript as only parasitological experimental data without bioinformatics, properly conducted, summarized, and discussed would be possible to consider in this form not suitable for publication. So much work put in for nothing, it's a shame.

I have many major comments which I couldn't show, too incorrect. I show only examples:

Abstract is not logical, and order presents the results and conclusion!!!!

Following your advice, abstract checked and corrected it

Line 21 -22 value I50 is important for results but here without data

Checked and added in line 15-16 and highlighted with yellow color.

Line 32-37 dose of Albendazole 100 ug/ml, next 500-200 ug/ml???, abbreviation ML, genes???

Concerning doses and abbreviation, it is checked and corrected in lines 17 and 25, highlighted with yellow colors. Concerning genes, according to your  opinion, we removed the bioinformatics study at all.

Introduction

Line 58 English needs corrected

Checked and corrected, highlighted with yellow color in line 49(in the recent copy after removal of bioinformatics study), the English of the whole manuscript was revised by a native English speaker

Line 79 and whole manuscript without corrected names of organism (big letters, italics….)

The whole manuscript is checked and corrected

Material and methods

Line 87 The T. spiralis strain was obtained as a muscle larvae rat (donor) from the Parasitol-87 ogy Department of the Biological unit of Theodor Bilharz Research Institute, Imbaba, 88 Giza, Egypt (TBRI). Is there approval of the ethics committee for rats???

The T. spiralis strain was obtained as a muscle larvae rat (donor) from the Parasitology Department of the Biological unit of Theodor Bilharz Research Institute, Imbaba, Giza, Egypt (TBRI) where all approved research work complied with the World Medical Associationof ethics under Fedral Wide Assurance No. FWA00010609.

We added this statement  in the manuscript

Line 148 it is a result not methods with figure

Checked and corrected.  

Line 163, bioinformatic methods is not correct without experiments analysis

 According to your opinion, we removed the bioinformatics study at all.

Discussion

There isn’t discussion, without name gens expansions of abbreviations etc……..

According to your opinion, we removed the bioinformatics study at all.

Line 361, bibliography without citation in the list

We checked the sentence and corrected it

Reviewer 3 Report

This study of the effects of eugenol on adults and larvae of T. spiralis has been well conducted, but the text requires considerable work to clean up the linguistic problems. It took me several reads in some places in the Methods to work out what was being done.

1. I do not like the use of %concentrations and if the concentrations were actually produced as micrograms/ml then that is what should be used.

2. I do not think that the bioinformatics material contained in 2.7 (methods) and 3.2 (Results) provide any helpful information and is in any case not discussed in any detail. If the paper is published, then it should be without the bioinformatics material

3. I would include some information on the Trichinella life cycle since this may not be familiar to all readers.

Author Response

This study of the effects of eugenol on adults and larvae of T. spiralis has been well conducted, but the text requires considerable work to clean up the linguistic problems. It took me several reads in some places in the Methods to work out what was being done.

  1. I do not like the use of %concentrations and if the concentrations were actually produced as micrograms/ml then that is what should be used.

Thank you for this sciemntific note,

Unfortunately in this study we used the % concentrations and perform the statistical analysis based on it, but in future studies, we intend to use the ug/ml measurement, thank you again for guiding us to a more accurate scientific measurement system.

  1. I do not think that the bioinformatics material contained in 2.7 (methods) and 3.2 (Results) provide any helpful information and is in any case not discussed in any detail. If the paper is published, then it should be without the bioinformatics material

According to your scientific advice, we removed the bioinformatics study at all

  1. I would include some information on the Trichinella life cycle since this may not be familiar to all readers

Following your instructions, we added this part in the section of introduction and highlighted it  with blue color from line .

Round 2

Reviewer 2 Report

I appreciate the improvement of the manuscript, but I still see the disorder and the lack of attention to detail in the preparation of the publication.

For example, Citations line 287 at the beginning of a paragraph.etc.

please use uniform concentration determination either percentage or ug/ml.

The IC50 is a results chapter, not material and methods. Here the molarity of the uM concentration is used, it would be necessary to replace everything (whole experimental methods with uM) to this concentration (instead of ug/ml change to uM).

In my opinion, publication is possible, but after a thorough editorial improvement.

Author Response

thank you for your scientific revision, we do all the required corrections

Reviewer 3 Report

Thank you for responding to my comments and those of the other reviewer. I am satisfied with the responses that you have given and the changes made. Inserting new material and deleting others often introduces additional problems with the layout and language. There is still a need to deal with typos and grammatical issues, but these should be straightforward. 

Author Response

thank you for your scientific revision, we did all the required revision
